# The Precision Analysis of Cutting Edge Preparation on CBN Cutting Inserts Using Rotary Ultrasonic Machining

**DOI:** 10.3390/mi13101562

**Published:** 2022-09-21

**Authors:** Marcel Kuruc, Tomáš Vopát, Jana Moravčíková, Ján Milde

**Affiliations:** Faculty of Materials Science and Technology in Trnava, Institute of Production Technologies, Slovak University of Technology in Bratislava, 81243 Bratislava, Slovakia

**Keywords:** cutting edge preparation, rotary ultrasonic machining, cubic boron nitride, ultrasonic assisted machining, profile measurement

## Abstract

This paper is focused on the issue of preparing the cutting edge microgeometry of cutting inserts made of cubic boron nitride (CBN). The aim of this research was to investigate the possibilities of rotary ultrasonic machining (RUM) for preparing asymmetric cutting edge microgeometries of various shapes (chamfers, circular, and elliptical rounding and their combinations) on a CBN cutting tool. In this article, a new type of advanced cutting edge preparation method is presented. CBN is relatively resistant to the most often used (abrasive) methods of cutting-edge preparation, due to its very high hardness (which is a prerequisite property for machining difficult-to-cut materials). Such hard materials could be processed using advanced manufacturing methods, and rotary ultrasonic machining (RUM) is one such method. Experiments have shown that RUM can be used for machining CBN. However, high hardness is not the only challenge here. For cutting edge preparation, it is necessary to achieve an adequate accuracy of size and dimensions. The presented paper analyzes the suitability of the RUM process for processing CBN inserts. The results of the experiment showed that this method can be used for preparing asymmetric cutting edge microgeometries with various shapes.

## 1. Introduction

As productivity is one of the most important aspects of industry there are many ways to increase it. For example, cutting tools made of advanced materials [1] or coated with advanced coatings [2] could be used. They could have improved geometry [3], or hybrid processes could be used, such as laser-assisted machining [4] or ultrasonic-assisted machining [5], etc. One of the least cost-demanding methods is cutting edge preparation [6], which needs neither tool material replacement nor additional coatings (although coatings are recommended), nor machine tool modifications, but can reduce cutting forces, mechanical stress, cutting temperature, surface roughness, tool wear, etc. [7,8]. Cutting edge preparation is based on the controlled creation of “tool blunt”: fillets and chamfers (and a combination thereof). Some examples of cutting edge preparation are shown in Figure 1. To achieve the required shape, several methods have been developed, such as micro blasting (dry or wet) [9], brushing [10], drag-finishing [11], magnetic abrasive machining [12], abrasive flow machining [8], etc. Besides these abrasive processes, in which productivity depends on substrate hardness, there are also advanced methods such as electro-erosion honing [13] and laser marking [14]. In addition, brand new processes are still being developing, such as cutting edge preparation using plasma discharges in electrolyte [15].

In particular, abrasive methods have difficulties in processing hard materials such as cubic boron nitride (CBN). CBN is the second hardest known material (diamond is first). Its occurrence is not natural; it is made artificially by a similar process as that for synthetic diamond creation. This means the high temperatures (1500 to 3230 °C) and high pressures (4 to 18 GPa) are maintained for a long time period [17,18]. Unlike diamonds, CBN has high chemical and thermal stability and no affection for iron. Therefore, it is suitable as a cutting material for heavy-duty applications for machining iron-based and difficult-to-cut materials [19,20,21]. CBN can reach a hardness of 70 GPa, and is thermally stable up to 1300 °C in the air (in a vacuum even higher) [22,23]. Figure 2 shows an application of CBN inserts for milling high chromium cast iron.

As CBN´s properties make it almost impossible to machine using conventional methods, advanced machining methods have been developed. For processing very hard and brittle materials, such as optical glass and technical ceramics, rotary ultrasonic machining (RUM) wasdeveloped. This machining method utilizes a diamond tool, which rotates around its axis (like a conventional milling cutter) and also vibrates along this axis in an ultrasonic frequency. The combination of those movements enables low cutting forces, mechanical stress, cutting temperature, surface roughness, tool wear, etc. to be achieved. The process is enhanced by a coolant, which also cleans the cutting tool [25,26,27]. The machine tool may appear similar to a traditional milling center (as shown in Figure 3a), but the ultrasonic cutter resembles a small grinding wheel (as shown in Figure 3b).

The progress of rotary ultrasonic machining for CBN is in its early stages. Usually, CBN is used as a cutting material. There are very few research works where CBN was used as a workpiece; mainly by the authors of this paper (Kuruc et al.). During the machining of CBN by RUM, a low surface roughness parameter Ra 0.24 µm was achieved [27]. The machine load (206–257 N), as well as torque (1.13–1.29 Nm), were increased with increasing cutting speed, feed rate, and depth of cut, while cutting speed and feed rate had only a small influence on machine loads [26]. Rapid tool wear was observed [28]; the grinding ratio achieved a value of only 2.7, which is approx. 26-times less in comparison with RUM alumina ceramic [16].

This is one of the first papers where RUM was used for cutting edge preparation. The presented research is built on previous research published in [16], where the research was focused on the possibility of processing CBN cutting inserts with a simplified shape (tip radius was ignored, and only chamfers were produced).

## 2. Materials and Methods

In the experiments, CBN cutting inserts provided by the company Halnn Superhard Tools (Zhengzhou, Henan Province, China) were used and clamped in the fixture as shown in Figure 3b. They had dimensions of 12.7 × 12.7 × 4.8 mm and crated with asymmetrical chamfers measuring 0.3 mm × 20° on one side (all four edges at once) and 0.6 mm × 20° on the other side Those chamfers were manufactured using a single tool path via continuous five-axes machining. Besides chamfers, fillets were also made. Elliptical fillets were created with dimensions of 0.25 × 0.1 mm on one side and 0.5 × 0.2 mm on the other side. Those fillets were manufactured using several tool paths via conventional three-axes machining, so the chamfers and fillets had similar sizes, and the second sides had double the size of the first sides. These cutting edge preparations are illustrated in Figure 4. Throughout the machining process, a coolant (oil emulsion) was mostly used for tool cleaning; no elevated temperatures were expected.

Cutting edge preparations usually do not achieve such high values. There are two reasons why they were chosen: #1 the cutting inserts had already had some initial cutting-edge preparation; #2 higher values often have a greater positive impact on the cutting process, as shown in Figure 5. It can be seen that the highest fillet of the cutting-edge caused the lowest tool wear, surface roughness, and cutting force. These benefits are maintained for the whole machining process [28]. However, increasing the cutting edge preparation size using the standard abrasive method would require a longer processing time (especially for harder specimens), and the process of cutting edge preparation could start to lose its profitability. The RUM process is not affected by this as much as other methods.

The production of the cutting-edge on CBN inserts was performed on a five-axes ultrasonic milling center DMG Mori Ultrasonic 20 linear (DMG Mori, Stipshausen, Germany) (previously shown in Figure 3a). In this machine tool, an ultrasonic tool-holder with an ultrasonic milling cutter Schott 6A9-Da24-2-6-14h6x8,4 (Schott Diamantwerkzeuge, Stadtoldendorf, Germany) was mounted. This cutting tool has a diameter of 24 mm, and contains diamond particles with an average size of 91 µm. In Table 1, the used machining parameters are recorded. These parameters were selected according to the manufacturer´s recommendations and our own experience. The tool vibration mode was set as one-dimensional longitudinal ultrasonic vibration along the vertical axis.

When the CBN cutting inserts were machined, their measurement began, with the use of the Surfcom 5000 surface texture and contour measuring instrument. This flexible device is produced by Accretech (Coventry, United Kingdom) and its very high precision of about ±0.2 μm means it´s often used in the automotive, aerospace, marine, medical, and mechanical industries. The cutting edge profile was measured in several places around the cutting insert. The device and measuring process are shown in Figure 6.

The samples were appropriately cleaned before the measuring process, and the recommended measuring conditions were set, such as measuring speed of 0.15 mm/s and probe radius of 0.002 mm. Stylus movement during the measurement was performed in the x-axis direction. Measurement length was 7 mm, as shown in Figure 6b. Measurement of cutting edge shape was carried out in the direction from the flank surface to the rake face of the cutting insert.

## 3. Results and Discussion

Measurement results are provided as multi-page reports, hence only some representative images and arithmetical means have been selected. Figure 7 shows the results of creating asymmetrical chamfers of 0.3 mm × 20° (Figure 7a) and 0.6 mm × 20°. The rake surface is located on the horizontal (X) line, and the flank (clearance) surface is located on the vertical (Y) line. The required values (0.3 and 0.6 mm) were measured in the horizontal direction and the required angle (20°) was measured between the horizontal line and the cutting edge surface.

Similar results were achieved for the elliptical roundings 0.25 × 0.1 mm (Figure 8a) and 0.5 × 0.2 mm (Figure 8b). However, higher inaccuracy can be seen. The dimensions of the major semi-axis of the ellipse (0.25 and 0.5 mm) were measured on the horizontal axis and the dimensions of the minor semi-axis of the ellipse (0.1 and 0.2 mm) were measured on the vertical axis.

The obtained values are recorded in Table 2 and Table 3. Differences in values between the cutting edges can be seen. This is caused by the demand for high precision in cutting insert dimensions and the placement of a zero point for the manufacturing process. Even a small inaccuracy is manifested as a big difference in the dimensions of cutting edge preparations between cutting edges.

A relatively high inaccuracy can be seen, especially for the angle. This could have been caused by the lower positioning precision of the machine tool cradle in comparison with the positioning of the translational axes. For easier interpretation of the results, the average values and standard deviations are recorded in Table 4. In Table 5, these values are recalculated into percentages.

The results recorded in Table 5 are visualized in Figure 9. The blue color represents the mean first value (in percentages), and the orange color represents the mean second value (also in percentages).

According to the results as recorded in Table 5 and visualized in Figure 9, it can be seen that the mean obtained values did not differ by more than 10%. The second mean values usually have a worse standard deviation. The reasons for this inaccuracy are described in more detail in below. The mean values represent the precision of the process, and the standard deviation represents its repeatability.

Besides the debatable achieved precision and repeatability of the manufactured cutting-edge preparations, the controlled creation of chamfers and fillets for CBN cutting inserts could be considered a success.

According to the results summarized in Table 4, a high standard deviation in chamfer angle for the first side of the first cutting inserts can be seen. This was caused by the alignment of the cutting insert in the fixture not being sufficiently precise.

The highest inaccuracy was for the angle, but on the second side of the first cutting insert. The cradle positioning was not as precise as the positioning in the translation axes. Moreover, a bigger chamfer was created on the second side, which means more workpiece material to resist, which causes a high torque for the cutting tool and higher tool wear. The cutting tool was in contact with the workpiece on its lateral side, where the tool is not as stiff as during machining using a rake surface.

The average linear parameters differed by 0.23 to 6.04% in comparison to the required values, which could be considered an acceptable precision in some cases. The worst average angular parameter differed by 13.65%, which would require improvement in most cases.

Rotary ultrasonic machining is an advanced technology developed for machining very hard and brittle materials. The experiment´s results showed wide scope for improvement, which will be an issue for further research. The results of the presented research can be summarized as follows:

When creating a cutting edge using rotary ultrasonic machining:The cutting edge preparation can have many shapes. Most of the methods are limited to only one shape (usually circular rounding). However, almost any shape can be created using three-axis machining.During three-axis machining, the cutting tool´s rake surface produces the cutting edge. The ultrasonic vibrations are perpendicular to the machined surface. There are several tool paths needed, and the resultant machined surface contains very small “cascades.” Although more tool paths cause a smoother surface, the machining time will be longer.The process of cutting edge preparation is often time-consuming (even several hours long). However, a chamfer around the whole cutting insert (one side) can be created in a couple of seconds when five-axis machining is used. This is caused by creating a cutting edge using a single tool path. Therefore, the processing time is very little affected by size.Five-axis machining can only create chamfers; however, with different sizes and angles.During five-axis machining, the lateral surface of the cutting tool produces the cutting edge. The ultrasonic vibrations are parallel to the machined surface and the cutting tool is more affected by the torque.CBN is very hard and abrasive. It can wear the lateral surface of the cutting tool during five-axis machining, which decreases the shape and dimensional accuracy.

A relatively high standard deviation could be affected by:The true dimensions of the cutting inserts differing from those written on the inserts box. Therefore, the tool paths that were generated in CAM according to the ideal 3D CAD model are displaced, and the tool´s relative position is changed. This could be solved by the very precise measurement of every single cutting insert before its processing.The zero point determined by a tough probe in the machine tool differs from that created in CAM software. The cutting insert is small and there is little space for tough probe movement. Moreover, the tough probe itself may not have been satisfactorily calibrated. Even a small displacement of the zero point from the cutting insert’s center would cause an increase in the cutting edge preparation on one side, and a decrease on the opposite side. This could be solved by applying a specially designed fixture.The zero point is determined with translation axes. However, an inclination in the rotational axes could negatively affect the accuracy. Cutting inserts have very small dimensions and it is difficult to measure their inclination. It is recommended to re-measure the inclination for every cutting insert before machining. This could also be solved by applying a specially designed fixture.Rapid tool wear causes changes in the cutting tool´s dimensions during the process. A harder workpiece material causes more intensive tool wear. Moreover, CBN is one of the hardest known materials. Tool wear cannot be prevented, and even its reduction could be problematic (the cutting conditions may already have been optimized). However, the effect of tool wear could be reduced if the tool is sent for re-measurement between processes. This could be reduced even more if the machining is repeated once more after re-measurement.The positioning precision of the machine tool itself has a major impact on determining the zero point, as well as the tool path coordinates. A low machine tool precision could be difficult (and expensive) to solve, but one cheap solution could be precision positioning measurement using an external device and consequently software correction of the positioning via a control system.

Relatively low precision could be affected by:Unsatisfactory positioning precision of the machine tool. A machine tool could lose accuracy due to aging and collisions. This could be reduced by positioning precision measurement using an external device, and consequently the software correction of the positioning via a control system.Differences between the cutting tool dimensions recorded in the machine tool and those used in CAM software can cause the wrong tool path to be generated. In CAM software, the real dimensions of the tool must be used rather than those indicated on the tool box.Difficulties of cutting tool measurement using a laser probe. Differences between cutting tool dimensions recorded in the machine tool and reality will manifest as inaccuracies. This would be reduced by the proper cleaning of the cutting tool and the calibration of the laser probe.Over time even the thermal expansion of the tool and workpiece materials could affect precision, if the machine tool cannot compensate for this. However, proper cooling could reduce this effect.Moreover, nothing is perfectly rigid. The machine tool has some stiffness. Even when only a very small amount of material is removed, the CBN material is very resistant. This could be reduced by using more tool paths during machining.

## 4. Conclusions

The controlled creation of cutting-edge preparation is a difficult process. Many processes can only create circular rounding, while the size is affected by the time of processing. Moreover, there can be differences between the edges of the cutting tool, or even along a single cutting edge. Some processes can be applied only on shank cutting tools made of HSS and carbide. Increasing the hardness of the tool makes the process even more difficult.

RUM technology has the potential to be used for cutting edge preparation. As this process was used for edge preparation for the first time, some difficulties can be expected. According to the results obtained, accuracy and repeatability are relatively good, considering the early (untuned) production process. Such technology can achieve high productivity, especially with five-axis machining. However, only a single cutting insert can be processed at a time. Asymmetric cutting edge microgeometries with different shapes, such as chamfers, circular and elliptical roundings, and their combinations, were created. The deviations between the required and manufactured dimensions and angles were calculated.

The technology´s precision is a very important factor, and must be improved before mass application in the tool industry. Low accuracy will cause higher differences between edges, as well as along the edge. When applied, this will be manifested as a less stable machining process, which could cause lower surface quality and shorter tool life.

In further research, the influence of the process parameters (cutting parameters, measurement of the cutting tool and cutting insert on the machining center, toolpath generation, NC program) on the accuracy of manufacturing will be determined. Moreover, a series of cutting experiments with a prepared CBN cutting tool is planned in order to demonstrate the effectiveness of the prepared CBN cutting tool.

## Figures and Tables

**Figure 1 micromachines-13-01562-f001:**
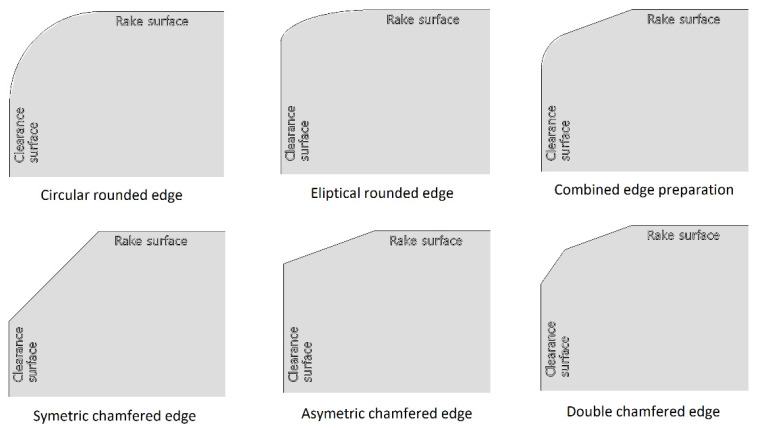
Typical geometries of cutting-edge preparation [16].

**Figure 2 micromachines-13-01562-f002:**
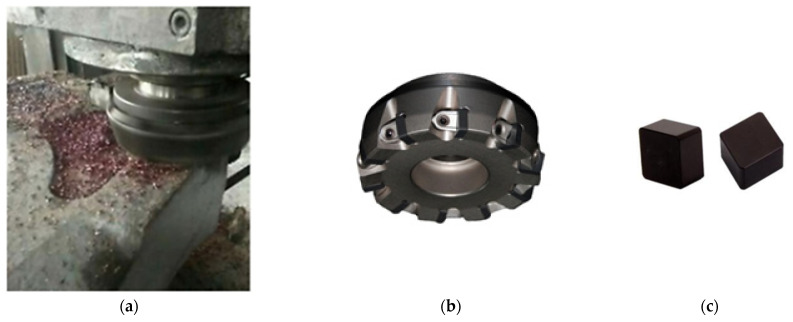
Application of CBN inserts: (**a**) milling process, (**b**) milling head, (**c**) CBN cutting inserts [24].

**Figure 3 micromachines-13-01562-f003:**
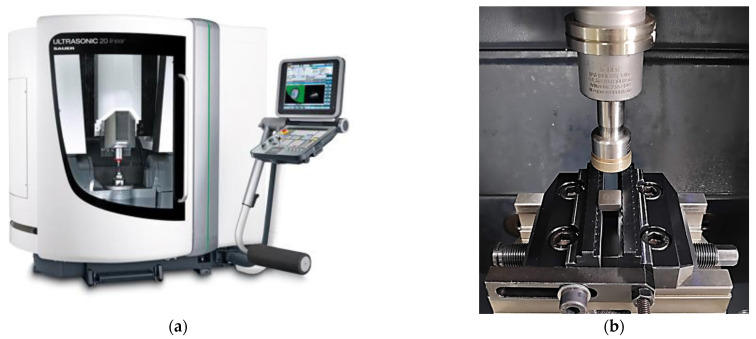
Rotary ultrasonic machining: (**a**) machine tool, (**b**) cutting tool [16].

**Figure 4 micromachines-13-01562-f004:**
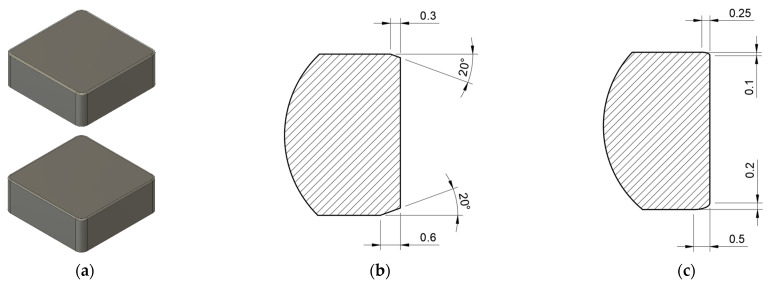
Cutting-edge preparations: (**a**) CBN cutting inserts, (**b**) asymmetrical chamfer, (**c**) elliptical rounding.

**Figure 5 micromachines-13-01562-f005:**
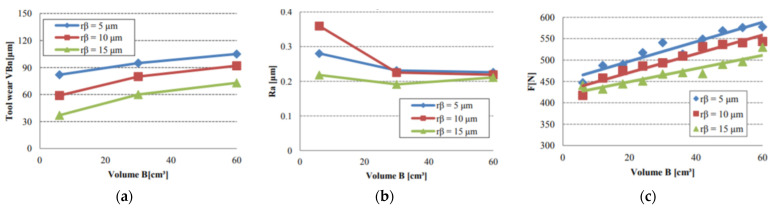
Influence of the cutting edge radius and machined volume on: (**a**) tool wear, (**b**) surface roughness, (**c**) cutting force [29].

**Figure 6 micromachines-13-01562-f006:**
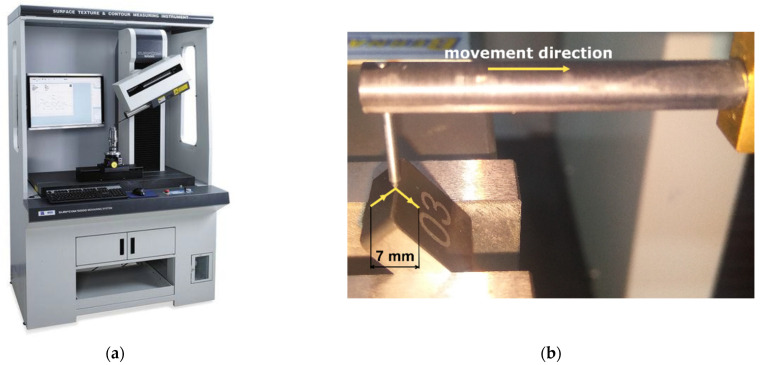
Contour measuring device: (**a**) Surfcom 5000, (**b**) measuring process [16].

**Figure 7 micromachines-13-01562-f007:**
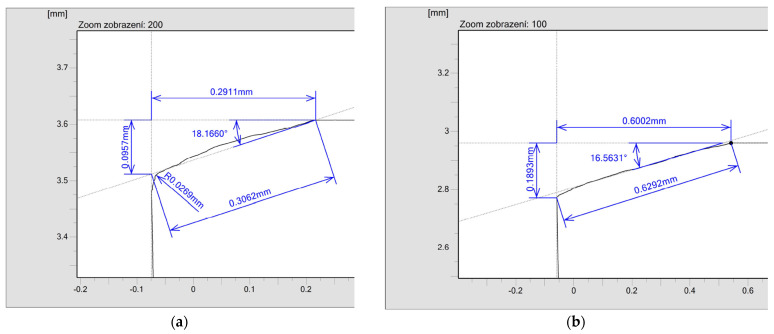
Dimensions of created asymmetrical chamfers: (**a**) 0.3 mm × 20°, (**b**) 0.6 mm × 20°.

**Figure 8 micromachines-13-01562-f008:**
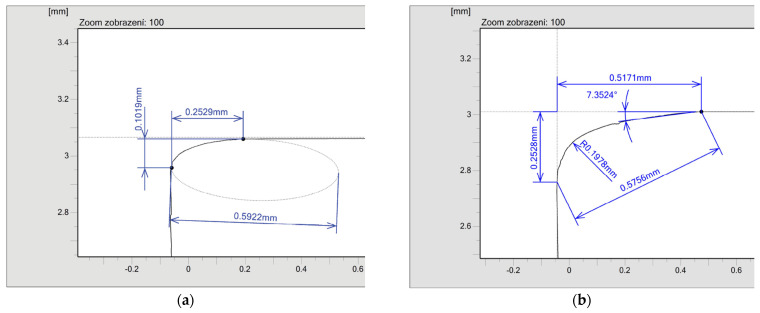
Dimensions of the created elliptical fillets: (**a**) 0.25 × 0.1 mm, (**b**) 0.5 × 0.2 mm.

**Figure 9 micromachines-13-01562-f009:**
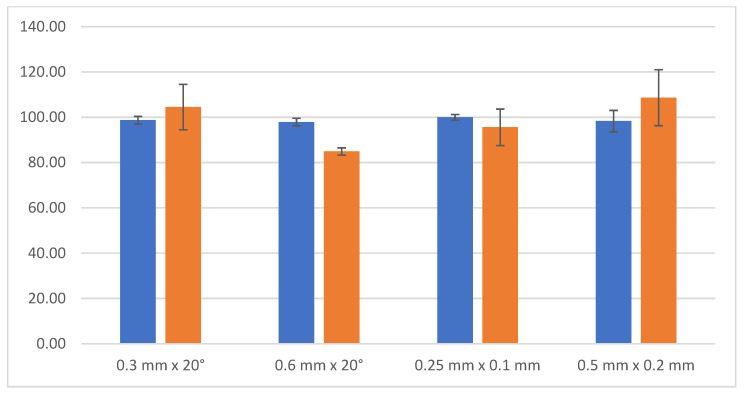
The percentage precision of the obtained cutting edge preparation.

**Table 1 micromachines-13-01562-t001:** Machining parameters used in the experiments.

Cutting Speed v_c_ (m/min)	Feed Rate v_f_ (mm/min)	Depth of Cut a_p_ (µm)	Ultrasonic	
			Frequency f (kHz)	Amplitude A (µm)
400	300	5	21.5	10

**Table 2 micromachines-13-01562-t002:** Required and obtained values for asymmetrical chamfers.

Edge	Required Dimension	Obtained Dimension	Required Angle	Obtained Angle
#1	0.3 mm	0.2911 mm	20°	18.1660°
#2		0.2973 mm		20.7369°
#3		0.2935 mm		22.7874°
#4		0.3027 mm		21.9188°
#5	0.6 mm	0.5805 mm	20°	16.9430°
#6		0.6002 mm		16.5631°
#7		0.5904 mm		17.3275°
#8		0.5778 mm		17.0797°

**Table 3 micromachines-13-01562-t003:** Required and obtained values for elliptical fillets.

Edge	Required Dimension	Obtained Dimension	Required Dimension	Obtained Dimension
#1	0.25 mm	0.2529 mm	0.1 mm	0.1019 mm
#2		0.2515 mm		0.0862 mm
#3		0.2455 mm		0.1027 mm
#4		0.2498 mm		0.0915 mm
#5	0.5 mm	0.4652 mm	0.2 mm	0.2026 mm
#6		0.5050 mm		0.2151 mm
#7		0.5171 mm		0.2528 mm
#8		0.4789 mm		0.1986 mm

**Table 4 micromachines-13-01562-t004:** The mean values of the obtained chamfers and fillets and their standard deviations.

Cutting-Edge Preparation	Mean First Value	Deviation of the First Value	Mean Second Value	Deviation of the Second Value
0.3 mm × 20°	0.296150 mm	0.005058	20.8998°	2.008721
0.6 mm × 20°	0.587225 mm	0.010206	16.9783°	0.319300
0.25 × 0.1 mm	0.249925 mm	0.003211	0.095575 mm	0.008068
0.5 × 0.2 mm	0.491550 mm	0.023721	0.217275 mm	0.024704

**Table 5 micromachines-13-01562-t005:** The mean values and their standard deviations in percentages.

Cutting-Edge Preparation	Mean First Value	Deviation of the First Value	Mean Second Value	Deviation of the Second Value
0.3 mm × 20°	98.717%	1.686%	104.499%	10.044%
0.6 mm × 20°	97.871%	1.701%	84.891%	1.597%
0.25 × 0.1 mm	99.970%	1.284%	95.575%	8.068%
0.5 × 0.2 mm	98.310%	4.744%	108.638%	12.352%

## Data Availability

The data presented in this study are available on request from the corresponding author.

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
