# Peer review of "The Precision Analysis of Cutting Edge Preparation on CBN Cutting Inserts Using Rotary Ultrasonic Machining"

_micromachines, 2022, doi:10.3390/mi13101562_

Round 1

Reviewer 1 Report

In this study, the rotary ultrasonic machining (RUM) was used to machining CBN. The results were greatly valuable. However, there are many problems to be improved.

1. In the introduction, the authors should point out the current progress of rotary ultrasonic machining for CBN. In addition, the innovations of this study should be presented.

2. In the materials and methods, the author should provide the detailed experimental setup.

3. The structure of this paper need to be modified. For example, the results and discussion should be combined, and the conclusions should be separately presented.

4. In the discussion, the reason for the improvement in accuracy during the RUM was insufficiently discussed.

5. The language of the manuscript is very poor. It needs a major revision.

Author Response

Thank you very much for your review and comments. We tried our best to include them in the manuscript at the appointed time. There are our answers:

1. Yes, we agree. The introduction of the paper was modified.
2. Thank you for this suggestion. Additional details have been added. If you are missing something specific, please write so.
3. Yes, we agree. The structure of the paper was modified.
4. Yes, we agree. The reasons have been added to the conclusion of the paper.
5. The article will be proofread in English by a native speaker. Probably by MDPI Language Editing Service.

Reviewer 2 Report

This paper is devoted to the precision analysis of cutting-edge preparation on CBN cutting inserts created by rotary ultrasonic machining. The results and discussion section are well argument and relevant for other researches working in the field. However, there are still a few points required to be modified, added or explained in more details, as follows.

(1) Reviewer found that most of the research works or images are from the reference [16]. Please point out the differences between this study and previously published reference.

(2) The key issue or innovation of the article is not stated in detail. For instance, a large section of the research background knowledge is introduced, while the correlation content of the research process or innovation results are not showed in abstract. The reviewer suggested rewriting or reorganizing the content of the abstract.

(3) The concept of rotary ultrasonic machining is relatively wide. Authors should point out the specific mode of vibration. one-dimensional longitudinal ultrasonic vibration, two-dimensional vibration? “Rotary ultrasonic machining” should be “Rotary ultrasonic assist machining”. Please refer to the literatureLiu Y, Li Q,N, Qi Z,C, et al. Defect suppression mechanism and experimental study on longitudinal torsional coupled rotary ultrasonic assisted drilling of CFRPs. Journal of Manufacturing Processes, 70, p.177-192.(2021).

(4) The description of the experiment process is not clear enough in part 3. Some details in Figure 6 need to be added.

(5) Some species after machining should be displayed in Figure 8. 9 for the comparison necessary dimensions.

(6) In part 4, it is necessary to introduce your specific research work. It is obviously not appropriate to introduce the background knowledge of rotary ultrasonic assist machining process again.

(7) The format of references needs to be uniform. For instance. references [1]-[5] are inconsistent with the others.

(8) The manuscript also required a spell/grammar check.

Author Response

Thank you very much for your review and comments. We tried our best to include them in the manuscript at the appointed time. There are our answers:

(1) Thank you for this suggestion. The differences were pointed out in the introduction of the paper.
(2) Yes, we agree. The structure of the abstract and paper was modified.
(3) Thank you for this suggestion. The mode of vibration was added in Chapter 2. But I am afraid, this technology is relatively young, and the terminology is not exactly established. The term “Rotary ultrasonic machining” is commonly used in the scientific community, e.g. by Z.J. Pei, R.P. Singh, M. Sundaram, W. Cong, and many more. The term “Ultrasonic assisted machining” usually refers to the machining process, where the common tools (like milling cutters) are used, but they are exposed to ultrasonic vibration. In our experiments, a special tool designed for this purpose was used.
(4) Thank you for this suggestion.  Fig. 6b was supplemented with the principle of measurement and measured length. In the text, a detailed description of measurement performing was inserted.
(5) I am sorry, but I don’t understand this comment of yours. Figures 7 and 8 displayed some species after machining including their necessary dimensions.
(6) Thank you for this suggestion. The specific research work has been added to the conclusion of the paper. Part 4 – “Discussion and Conclusions” introduced results achieved by RUM. I don’t think they should be considered as background knowledge.
(7) I am sorry, but I don’t see the difference. Could you be more specific, please? Maybe it is just an issue with your downloaded PDF. However, I copy the format from other references and use it on mentioned references.
(8) The article will be proofread in English by a native speaker. Probably by MDPI Language Editing Service.

Reviewer 3 Report

The application of RUM on the preparation of CBN cutting tool is a good topic, and it is meaningful to the industry.

But the current research in the manuscript  is not enough to demonstrate the pocessing effects of RUM.

(1) The manuscript just gives one set of the experimental data about the accuracy of the prepared cutting tool edge, Table 4 and Table 5, as well as Figure 9 contain almost the same information with Table2 and Table 3, the authors need to make it concise and remove the information redundancy.

(2) The research in the manuscript at least needs to cover the cutting force, surface morphology and diamond cutting tool wear of the RUM of CBN materials. These information will give a better understanding for RUM of CBN cutting tool.

(3) Futher, after preparing the CBN cutting tool, a series of cutting experiments should be designed and conducted to demonstrate the effectiveness of the prepared CBN cutting tool.

(4) In the introduction part, the author can remove the general figures such as Figure 2, 3, 5 to make the manuscript more concise.

Author Response

Thank you very much for your review and comments. We tried our best to include them in the manuscript at the appointed time. There are our answers:

(1) Thank you for this suggestion. Table 4 records results in absolute units to highlight the need for the measuring device with very high precision. Table 5 records results in relative units to simplify the comparison of the results. Figure 9 shows the results from Table 5 – in the table is the exact expression of the values with lesser informative value, and in the figure are more informative values with a lesser exact expression of the values. In Table 2 are recorded values for chamfers. In Table 3 are recorded values for fillets. Despite of similarity, I believe there is no information redundancy, only in Figure 9, but it has its reason.
(2) Yes, we agree. That information has been added in the introduction of the paper.
(3) Thank you for this suggestion. It is planned for future work.
(4) Thank you for this suggestion. Figure 3 and 6 shows the experimental set-up, which was required. Figure 5 explains the reasons for selected required values of cutting edge preparation. Therefore, I consider them as important.

Round 2

Reviewer 1 Report

  • It has been amended as requested and approved for publication.

Reviewer 2 Report

 The authors have addressed the concerns in a careful way. I recommend its publications.

Reviewer 3 Report

The research can go deeper in the further work